# Comprehensive Characterization of the Odor-Active Compounds in Different Processed Varieties of Yunnan White Tea (*Camellia sinensis*) by GC×GC-O-MS and Chemometrics

**DOI:** 10.3390/foods14020271

**Published:** 2025-01-15

**Authors:** Junaid Raza, Baosong Wang, Yue Duan, Huanlu Song, Ali Raza, Dongfeng Wang

**Affiliations:** 1Laboratory of Molecular Sensory Science, School of Food and Health, Beijing Technology and Business University, Beijing 100048, China; 2School of Exercise and Nutritional Sciences, San Diego State University, San Diego, CA 92182, USA; 3College of Food Science and Engineering, Ocean University of China, Qingdao 266003, China

**Keywords:** white tea aroma, shaking, sensory evaluation, odor-active compounds, GC×GC-O-MS

## Abstract

This study investigates the aroma characterization of unique white tea varieties from the Lüchun county of Yunnan province, Mainland China. These include shaken, unshaken, steam-cooked, and compressed varieties. The aroma profile of white tea varieties was analyzed using two-dimensional gas chromatography–olfactometry–mass spectrometry (GC×GC-O-MS), electronic nose (e-nose), and descriptive sensory evaluation. A chemometric approach was used to compare sensory scores to instrumental data. A total of 154 volatile compounds were detected in 16 white tea varieties through GC×GC-O-MS. Among these, 133 compounds were successfully identified through the National Institute of Standards and Technology (NIST) library, and 21 were listed as unknown. The identified volatile classes include aldehydes, such as hexanal and heptanal, which contribute to the green aroma of white tea, and alcohols like 2-heptanol and 3-hexen-1-ol, which exhibit fresh and floral odor notes. The content and relative odor active values (r-OAVs) of the volatile compounds were calculated. The chemometric data revealed significant variations in volatile contents between shaken and unshaken white tea varieties. The orthogonal partial least squares discriminant analysis (OPLS-DA) model showed strong validity and stability. This study describes the impact of processing conditions on the flavor profile of white tea and provides a solid foundation for monitoring the aroma quality of different processed white tea varieties.

## 1. Introduction

Tea (*Camellia sinensis*) is the second-most consumed beverage in the world, after water. Teas are generally classified as black tea, green tea, oolong tea, dark tea, yellow tea, and white tea, although the types of teas consumed globally also depend on several factors, including demographics, region, location, culture, and consumer acceptance and behavior [1,2]. The aroma quality of tea is largely dependent on some key factors, such as its production, plucking, processing conditions, oxidation, drying process, and storage conditions [3,4]. After withering, the shaking treatment also affects the quality parameters of white tea. Typically, white tea leaves are shaken for at least three to five cycles. It is still believed that the best varieties of tea are those that are prepared through careful handling by professional tea masters, from plucking to packaging, rather than through the use of machinery. Despite this, there are still key areas that need to be explored to achieve the best-quality teas, including their flavor and sensory profiles.

Known as one of the six types of tea, white tea is mainly produced in the Yunnan and Fujian provinces of China. White tea is minimally processed, and unlike other varieties of tea, it does not undergo complete enzyme inactivation or fermentation [1,5]. The production of white tea primarily involves the basic processes of withering and drying. Additionally, withering plays a very crucial role in determining the sensory quality of white tea [6,7]. A previous study reported that light-emitting diode (LED), specifically blue and red lights, used during the withering process of white tea significantly enhances its taste and aroma profile [8]. Shaking, on the other hand, is an optional technique that was originally adapted for oolong tea production and has recently been applied to other teas, including black tea, to improve its organoleptic properties [9,10]. The shaking process causes mechanical damage that promotes the accumulation of metabolites, contributing to the unique aroma of oolong tea [11]. Research on the effect of the shaking treatment on the aroma properties of white tea is limited. Previous research has shown that white tea varieties from the Yunnan, Xinjiang, and Fujian production regions can be distinguished based on their sensory profiles and compound differences, indicating that classifying white tea by production region is both possible and impactful [12]. Major varieties of white tea include Yin Zhen Bai Hao (silver needle, one of the high-quality white teas), Bai-Mudan (white peony), Gong Mei (tribute eyebrow), and Shou Mei (long life eyebrow variety) [13]. Unlike other kinds of teas, white tea is slightly fermented, and only two major steps (withering and drying) are involved in its processing [14].

As the consumption of white tea around the world has grown, many studies have focused on its aroma properties. White tea primarily contains floral, fresh, fruity, woody, sweet, and caramel-like aroma notes [4]. A recent study identified 10 key aroma-active compounds that significantly contributed to the sweet and fruity aroma of Jinggu white tea, with benzeneacetaldehyde and linalool being the main aroma contributors [15]. The primary volatile classes in white tea include alkanes, aldehydes, ketones, alcohols, sulfur-containing compounds, esters, heterocyclic compounds, and aromatics, with alcohols and aldehydes comprising over 60% of the odor profile, thus contributing the most to its fresh and green aroma [16]. Hexanal has been frequently reported in many white tea aroma studies and is associated with the fresh, green, and grassy odor of white tea [1,17,18]. White tea does not contain a significant amount of Maillard reaction products or an abundance of methoxy-phenolic compounds [7]. Studies have shown that the majority of the aroma compounds in white tea are derived from glycoside hydrolysis and fatty acid degradation [19].

Headspace solid-phase microextraction (HS-SPME), in combination with two-dimensional gas chromatography–olfactometry–mass spectrometry (GC×GC-O-MS), is a remarkable and novel technique for the identification of trace components of tea samples [1,20]. Conventional gas chromatography–mass spectrometry (GCMS) is commonly used to identify volatile compounds in teas. Recently, techniques like GC×GC-O-MS and HS-SPME-GC-IMS have been applied to quantify aroma classes in 18 different tea types [12].

The application of GC×GC-O-MS in food is remarkable due to its sensitivity, high resolution, and concentration capacity. These features enable complete separation, which is not possible with one-dimensional GC systems. The most important aromatic compounds previously analyzed in the two white tea varieties (Silver Needle and White peony) using HS-SPME-GCMS were β-linalool, phenylethyl alcohol, and geraniol, while ketones, aldehydes, and hydrocarbons were less abundant [21]. A systematic investigation into the aroma formation of white tea identified 172 volatile compounds and showed that glycosidically bound volatile compounds and free aroma precursor amino acids are the main contributors to the unique aroma of white tea [6]. Some researchers recently used HS-SPME-GCMS analysis on a newly discovered white tea variety (locally known as Caicha); the key odorant classes in their study comprised a substantial amount of terpenoids, alcohols, aldehydes, ketones, and some esters, with important volatiles including α-phellandrene, heptyl ester 2-methyl-propanoic acid, 3,5,5-trimethyl-3-cyclohexen-1-one, and 3-methyl-benzaldehyde [22]. The effect of tea processing treatments on its aroma profile can be identified using the GCMS technique. Gas chromatography–mass spectrometry (GCMS) has previously been used to analyze acrylamide in tea samples; for example, a study found that a higher (≥120 °C) roasting temperature was associated with the production of acrylamide [23]. Research on the impact of the shaking treatment on the relationship between the volatile compounds and catechin profile of oolong tea has revealed a significant correlation between the shaking process and the transformation of tea composition. Studies have also shown that shaking the fermented tea leaves improves its smell and odor profile [24,25].

The Lüchun county of the Yunnan province of China is famous for producing special white tea varieties; however, limited research has been performed on such commercial varieties. This study used a molecular sensory science approach, in combination with multivariate statistical analysis, to analyze the aroma profile of special white tea varieties. A total of sixteen different commercially processed white tea varieties from the Luchun county of Yunnan province of China were collected from different tea processors, and their volatile organic compounds (VOCs) were characterized using GC×GC-O-MS and chemometric analysis. The human sensory evaluation was compared to the instrumental odor evaluation through an electronic nose (e-nose). Moreover, a clustered heat map analysis of the chromatography results of all white tea varieties was performed. Subsequently, orthogonal partial least squares discriminant analysis (OPLS-DA) and principal component analysis (PCA) were performed to determine the key differences between different white tea varieties.

## 2. Material and Methods

### 2.1. Chemicals and Reagents

The chemicals and reagents used in this study were procured from multiple suppliers. The internal standard 2-methyl-3-heptanone (GC grade) and N-alkanes (C_7_-C_25_) were purchased from Sigma-Aldrich Co, Ltd. (Beijing, China), and n-Hexane (GC grade) was procured from Fisher Chemicals (Shanghai, China). Ultra-high-purity helium gas (>99% purity) and nitrogen gas (>99% purity) were acquired from Beijing AP BAIF Gases Industry Co., Ltd. (Beijing, China). Sodium chloride (NaCl), with 99.5% purity, was procured from Sinopharm Chemical Reagent Co., Ltd. (Shanghai, China).

### 2.2. Pre-Treatment of Yunnan High-Aroma White Tea

A total of sixteen different white tea varieties were used in this study and were procured from different tea processors of the Lüchun county of Yunnan province of Mainland China. A schematic flowchart of the methodology used in this study is presented in Figure 1. The white tea samples were processed by professional tea masters using different conventional tea processing techniques. After careful hand plucking, the white tea leaves were subjected to withering, shaking, drying, steam cooking, compressing, sorting, and grading treatments by professional tea manufacturers. To reduce the chances of errors, the collection of the samples was randomized, and the samples were collected from different processed batches. Before the analysis, each sample set was ground for at least 10 s, which was repeated three times, to make fine tea powder for later use by using a laboratory-scale LG-01 grinder (Baixin Pharmaceutical Machinery Co., Ltd., Wenzhou, China). The samples labeled SS1 to SS7, NSS1 to NSS7, OUS, and OUC (Figure 2A–C) represent the varieties prepared through traditional processing methods and subjected to shaking, no shaking, and other treatments, respectively. The origin, detailed information, processing methods, and shaking times for each white tea variety have been explained in Figure 2A–C. To store these samples, the room temperature was maintained at 25~26 °C and the humidity level was set to 70~75%. Freshly procured white tea samples were then subjected to sensory and instrumental analyses.

### 2.3. Preparation of White Tea Infusions

Three different tea infusion methods were initially prepared for the instrumental analysis of high-aroma white tea samples in agreement with the Chinese National Standard for the Sensory Evaluation of Tea (GB/T23776-2018) [26]. These include tea infusions prepared with white tea leaves and brewed with boiling water at 100 °C, tea infusion with tea powder and brewed with boiling water at 100 °C, and tea infusion with tea powder and brewed at room temperature (25 °C). After comparing all three tea infusion pre-treatments through sensory evaluation, the tea infusion prepared with tea powder at room temperature was selected for instrumental analysis. For that, raw white tea leaves were ground with a laboratory-scale LG-01 grinder (Baixin Pharmaceutical Machinery Co., Ltd., Wenzhou, China), and then, 5 mL ultra-pure water was poured into 0.5 g of white tea powder [26]. The samples were covered with lids for 3 min, and the tea infusions were transferred to headspace vials for further analysis. The same method was applied to all other tea infusions mentioned in Section 2.4 and Section 2.6.

### 2.4. Sensory Evaluation of Sixteen Different Varieties of Yunnan White Tea

Comprehensive sensory evaluation is particularly important for consumer acceptance of white tea. A descriptive sensory evaluation approach was applied to evaluate the odor descriptors of different varieties of white tea. For the sensory evaluation of white tea samples, initial recruitment was performed, and sixteen sensory panelists (8 females and 8 males) with an average age of 26 years were recruited from Beijing Technology and Business University. First, different flavor attributes were described to the sensory panel after a comprehensive discussion, and with the consent of all the panelists, some important/potent attributes were selected for the sensory evaluation. Tea sensory wheels from a previous study [27] were used while presenting the tea samples to the panel, and after a detailed sensory discussion, the sensory liaison for Yunnan high-aroma white teas was obtained. These descriptors include woody, floral, fruity, nutty, grassy/fresh leaf, and herbaceous aromas. The definitions for each descriptor and the reference compound used for their training are presented in Table 1. A pre-screening sensory test of the panel was performed prior to the final sensory evaluation for the descriptors. For the pre-screening test, three concentrations were prepared for each descriptor (slight, medium, and intense), and participants were only allowed to participate in the final sensory evaluation if they were able to pass the pre-screening test. The samples for the final sensory analysis were prepared using the method mentioned in Section 2.3. Freshly prepared tea infusions were transferred to tea cups and kept at 25 °C for sensory evaluation.

The samples were randomized, labeled with random codes to avoid any biased data, and transferred to amber vials prior to analysis. All the information regarding the samples was kept confidential during the sensory test to ensure the validity of the sensory evaluation. A 5-point (0–5) hedonic scale sensory evaluation procedure was performed in accordance with the China National Standard (GB/T 23776-2018). The panelists were carefully trained for each score prior to analysis. A score of 0 means that no odor was detected, a score of 1–2 means that a weak odor was detected, a score of 2–3 means that a medium/moderate odor was detected, and finally, a score of 4–5 means that an extremely strong or persistent odor was detected. The sensory evaluation was carried out blindly and independently by each panelist in separate, quiet, and odorless sensory booths for the purpose of avoiding any environmental and human interference during the sensory test. Each sample was prepared in triplicate and was sniffed and scored by the panelists three times; then, their mean scores were calculated.

### 2.5. E-Nose Analysis

An electronic nose (e-nose) instrument (PEN3, Airsense Analytics GmbH, Schwerin, Germany) was used to analyze the volatile compound classes of different white tea varieties. Before analysis, the e-nose was carefully calibrated, the flushing time for the analysis was 190 s, the detection time was set to 200 s, and the measurement interval time was 1 s. The ten sensors used in the e-nose system and their characteristic sensitivities toward the classes of volatile organic compounds are described in Section 3.2. The chamber and purge flow rates were set to 300 mL/min. The white tea sample (0.5 g) and 5 mL of ultra-pure water were transferred into a 20 mL headspace vial for analysis. The samples were pre-heated in a water bath at 55 °C for approx. 20 min and subjected to e-nose analysis immediately afterward. The samples were assessed six times for accuracy.

### 2.6. HS-SPME for the Extraction of Odor-Active Compounds in White Tea

For the headspace extraction of volatile compounds, the tea infusions were prepared according to the method described in Section 2.3. Freshly brewed tea infusions were prepared (tea/water ratio of 1:10) and subjected to a wait time of 3 min by covering the samples with lids prior to transferring them to SPME vials. NaCl (0.25 g) was added to each vial to facilitate the volatile extraction. For the quantification of volatile compounds, 1 μL of internal standard (2-methyl-3-heptanone) with a concentration of 0.816 μg/μL was added to each SPME vial. A special method was optimized for HS-SPME. For aroma extraction, a 2 cm divinylbenzene/carboxen/polydimethylsiloxane (DVB/CAR/PDMS)-coated SPME fiber (50/30 μm, Supelco, Bellefonte, PA, USA) was used [28]. The samples were pre-heated at 60 °C for 20 min. The final aroma extraction was performed at 60 °C for 40 min. The SPME fibers were pre-conditioned at 230 °C for at least 5 min prior to final analysis, as recommended by the manufacturer. The desorption of the analytes through SPME fibers was then performed for 6 min in the back inlet of the GC-O-MS instrument.

### 2.7. Comprehensive Two-Dimensional GC×GC-O-MS Analysis

For the identification of volatile compounds, we used an Agilent GC system (8890) connected to a mass detection system (5977B), Agilent Technologies, Santa Clara, CA, USA. The instrument was also connected to an olfactometry detection port (ODP-3), Gerstel, Germany. The separation of analytes in white tea samples were performed on two different columns, a DB-Wax capillary column (36 mm × 250 μm × 0.25 μm film thickness) and a mid-polarity secondary column DB-17MS (1.9 mm × 180 μm × 0.18 μm), Agilent Technologies, Beijing, China. A special GC×GC–O–MS method was customized by adapting the protocol mentioned by Yang et al. [29] as needed. A solid-state modulator SSM1800 (J & X Technologies, Shanghai, China) was installed to regulate the heating and cooling phases between the two columns. The temperature of the cold zone was set to −50 °C, and the temperature of the heating zone was set to 70 °C. The outlet zone’s heating temperature was set to 160 °C, and the modulation period was set to 4 s. The GC×GC oven temperature was set to 50 °C, slowly raised to 230 °C at 3 °C/min, and then held for 5 min. The temperature of the back inlet was 230 °C. The flow rate of the carrier gas helium (99.999% purity) was 1.6 mL/min. A splitless injection mode was applied. The temperature of the ion source was set to 230 °C. The quadrupole temperature was adjusted to 150 °C. The temperature of the transmission line was adjusted to 280 °C. A full scan mode was applied for acquisition. For mass scanning, a range of 50–550 m/z was used. The electron impact ionization was set to 70 eV. The total GC×GC-O-MS run time was 58.33 min. To allow participants to sniff aroma compounds through the instrument, an olfactory detection port (ODP) was installed between the GC system and the MSD system. The operational parameters of the ODP were adjusted. The temperature of the ODP transmission line was set to 280 °C, and the temperature of the sniffing port was set to 200 °C. The split ratio between the olfactory detection port and the mass spectrometry detector (MSD) was 1:1. The ODP sniffing port was continuously ventilated with ultra-high-purity nitrogen gas. Moisture was continuously supplied through the ODP port to keep the sniffer’s nasal cavity moist and avoid the drying of nose mucus during the sniffing experiment. At least five trained sniffers (three females and two males) were recruited to evaluate the odor descriptors and their intensities from the ODP. These sniffers were provided with at least 90 h of sensory training. A series of pure standard compounds were used as reference material to train the sniffers. Only sniffers who were able to distinguish the different odors of the reference standard compounds were allowed to participate in the final experiment. The sniffers were shuffled during sniffing experiments to avoid fatigue. The odor properties and odor intensities of the individual aroma compounds were recorded by each sniffer on a scale of 1–4, where 1 means that a low odor was detected through the ODP, 2 means that a moderate odor was detected, 3 means that a strong odor was detected, and 4 means that a very strong and intense odor was detected. The time delays between an odor coming out of the ODP port and the retention times of the odorants shown in the real-time chromatograms were adjusted in the software after the analysis. If at least three sniffers were able to detect a similar odor on the same retention time (RT), then those odorants were selected and further identified through the MS library and online flavor and odor databases.

### 2.8. Qualitative Analysis of Aroma Compounds

The qualitative analysis was performed using Canvas GC×GC data processing software version V2.5.0.0. The signal-to-noise ratio for minimum peak detection was adjusted to 10. The compounds were tentatively identified by matching the mass spectrum of the unknown compound with those available in the offline National Institute of Standards and Technology (NIST) library and by matching the estimated retention index of the possible volatile compound with those available in the NIST online database. The peak areas of the samples were generated and compared using Canvas software (version V2.5.0.0). For compound discovery, the positive match was >700 and the reverse match was >800. The RI deviation was ≤30. For library searches, a minimum criterion of 85% was used to select a volatile compound. The linear retention indices (LRIs) of the unknown compounds were calculated using a standard mixture of n-alkanes (C_7_–C_25_) and were compared and matched with those reported in the literature. For that, 1 μL of a standard mixture of n-alkanes was injected into the GCMS instrument under the same chromatographic conditions, and the Kovats retention indices were calculated for each aroma compound. The compounds were also tentatively identified by matching the odor descriptors of the aroma compounds detected through the ODP. The odor descriptors were matched through online available databases like flavornet.org, the Good Scents Company database, odor and flavor detection thresholds by Leffingwell & Associates, Flavor DB, and the LRI and odor database. The linear retention indices (LRIs) of the unknown compounds were calculated using the following equation.$$RI=100N+100n\;[\frac{tr\;(x)-tr\;(n)}{tr(n+1)-tr\;(n)}]$$

Here, *RI* represents the retention index of a volatile compound, *N* represents the number of carbon atoms of the lower alkane, *n* indicates the difference between the carbon atoms of the two alkanes, *tr* is the retention time of an unknown volatile compound, *x* represents the retention time of the unknown compound that is being analyzed, *tr* (*n*) is the retention time for the lower alkane, and *tr*(*n*+1)is the retention time of the upper alkane [30]. The calculated RI was matched with the NIST-LRI, the Delta-LRI (Δ-LRI) value was calculated, and a cutoff value of (±30) was selected for the identification of the compounds.

### 2.9. Quantitative Analysis of Volatile Compounds

The quantification of volatile organic compounds in processed white tea varieties was performed using a semi-quantitative method (internal standardization methodology). The internal standard used was 2-methyl-3-heptanone, with a concentration of 0.816 μg/μL. The internal standard (1 μL) was added to freshly brewed white tea infusions (using a tea/water ratio of 1:10). The concentration of the unknown compounds was calculated using a formula from a previous study [31].$$C_{x}=\frac{C_{i}\times A_{x}\;}{A_{i}}$$

Here, *C_X_* is the concentration of unknown volatile compounds in white tea samples. *C_i_* is the concentration of the internal standard used for quantification, *A*_x_ is the peak area of the unknown volatile compound, and *A_i_* is the peak area of the internal standard.

### 2.10. Determination of Relative Odor-Active Values (r-OAVs)

The relative odor-active values (r-OAVs) of the aroma compounds in 16 different varieties of Yunnan high-aroma white teas were calculated by taking the ratio of the relative concentration of each identified aroma compound to its own odor threshold in water. The odor threshold of the aroma compounds was collected from previous studies, databases, and a book (compilations of odor threshold values in air, water, and other media) [32]. The r-OAVs of aroma compounds were calculated using the following equation.$$rOAV=\frac{C}{t}$$

Here, *C* represents the concentration of the aroma compounds, and *t* represents the odor threshold of an odorant in water.

### 2.11. Chemometric Data Analysis

Orthogonal partial least squares discriminant analysis (OPLS-DA) was performed using SIMCA (version 13.0) software (Simca Umetrics, Umeå, Sweden). For OPLS-DA model analysis, two types of scaling (Pareto (Par) and unit variance (UV)) were used for all variables in the dataset. The e-nose data were analyzed through principal component analysis (PCA). The contents of the volatile compounds in Yunnan white tea samples were also analyzed through hierarchical cluster heat map analysis (HCA). The heat maps were generated through TB Tools software (version v2.142). The sensory scores were analyzed by a spider plot generated through Origin Pro, version 2024, Origin Lab Corporation, Northampton, MA, USA. The contents of the volatile compounds were used to cluster different white tea samples. The univariate data analysis was performed through one-way ANOVA. A post hoc method was used in combination with a Duncan test to analyze the significant (*p* ≤ 0.05) differences among white tea samples. All the results in this study are expressed as the mean (by taking the average of replicates) and standard deviation (SD) values.

## 3. Results

### 3.1. Sensory Evaluation of Different Processed Varieties of White Tea

The quality of white tea is determined by its organoleptic and sensory properties. Figure 3A presents the sensory scores of 16 different varieties of Yunnan high-aroma white tea samples. White tea varieties processed with different shaking conditions, including shaking once, shaking twice, no shaking, steam cooking, cake form, and a combination of other conventional tea processing methods, were tested for sensory evaluation. Freshly brewed white tea samples were presented to the expert panelist for aroma evaluation. The aroma evaluation was conducted after comprehensive sensory training, as mentioned in Section 2.4. The sensory experts agreed that there was a large difference in the odor properties of the 16 different varieties of white tea. After the first discussion session of sensory training, the odor descriptors of the 16 Yunnan white tea varieties were described as woody, floral, fruity, nutty, grassy/fresh leaves, and herbaceous. Almost all samples exhibited a strong floral, herbal, or woody odor, which are mainly considered as potent aroma descriptors of white tea. The sensory results revealed that the highest woody (4.2), fruity (4.5), floral (4.5), and green (4.3) aroma scores were observed in OUC (Table 1) tea varieties. The highest herbal aroma score (4.53) was recorded in the OUS tea variety. The roasted aroma was highest (4.1) in SS6 (Figure 3A). The highest herbal score was observed for OUS (4.1). Overall, the lowest score was observed for the NSS7 white tea variety.

### 3.2. Results of E-Nose Analysis

Different white tea varieties were analyzed using a portable e-nose system (PEN 3, Airsense Analytics, GmbH, Sxhwerin, Germany). The e-nose system was equipped with an array of 10 sensors that are sensitive to different groups of volatile compounds. When the aromatic volatile compounds react with these sensors, electronic signals are generated based on the responses received from each signal. Figure 3B (e-nose radar chart) presents the aroma profile analysis of different white tea varieties from e-nose based on the responses received from 10 different sensors. W1C and W3C are the sensors specific to aromatic compounds, W5S represents the sensor dedicated to a broad range of volatile compounds, W6C represents the hydrogen sensor, W5C is the sensor for aliphatic compounds, W1S represents the sensor for methane compounds, which contains single carbon atoms attached to four hydrogen atoms, W1W is the sensor for sulfur-containing volatile organic compounds, W2S represents the sensor for alcoholic compounds, W2W shows the sensor for sulfur–chlorine compounds, and W3S represents the sensor for methane-aliphatic compounds. The highest signals were recorded from the W5S sensor, which exhibited sensitivity towards aromatic volatile organic compounds. W2S had the second-best signal response from the white tea samples, exhibiting sensitivity toward the alcoholic VOCs. The highest score recorded on the WS5 sensor was from the OUC sample, followed by OUS and SS6. However, the lowest responses recorded on the WS5 sensor were from NSS1 and NSS2. The highest scores recorded on the W2S sensor were from OUC, followed by SS4 and NSS3. The results revealed that the shaken, steam-cooked, and compressed white tea samples with higher contents of aromatic and alcoholic VOCs also obtained better responses from both W5S and WS2 sensors. NSS 1 had the lowest response from both sensors. A principal component analysis (PCA) was successfully applied as a chemometric approach to identify the differences or similarities among white tea samples and to determine the principal components. PCA is a technique based on linear dimensionality and has wide application in the analysis of complex sensory data. Figure 3C shows the PCA plot of the responses received from 10 e-nose sensors based on the score values. The figure clearly depicts that the PC1 component had a total contribution of 84.7%, while the contribution rate of PC2 was 5.4%. These findings reveal that the two principal components are able to explain most of the variation in the datasets. The PCA plots also showed overlapping responses in PC1 and strong proximity between the shaken varieties of white tea samples. A larger contribution rate reflects a better fit of the PC model and confirms the originality of the information, i.e., a significant variation between the contents of shaken and unshaken white tea varieties. This also explains that the white tea varieties, based on their shaking treatments, showed significant differences in their responses to aromatic and alcoholic VOCs. This could be because the shaking treatment accelerates the breakdown of the cellular structures, eventually increasing the oxidation process and allowing the release of aromatic and alcoholic compounds.

### 3.3. GC×GC-O-MS Analysis of the 16 Different Varieties of Yunnan White Tea

Appendix A present a total ion chromatogram plot of the sixteen different processed varieties of white tea obtained through 2D GC×GC-O-MS. A total of 98 aroma compounds were detected in the OUC variety, whereas a total of 117 aroma compounds were detected in NSS-1 Appendix A. This study successfully identified many potent odor-active compounds of white tea, including hexanal (delivering a grassy and tallow aroma), octanal (fruity, herbal, and green), nonanal (rose-like and floral), linalool (citrusy and musty aroma), phenylethyl alcohol (waxy and rosy), and (E)-beta-damascenone (aromas associated with apple and honey). Interestingly, the results revealed that the shaken, steam-cooked, and compressed white tea varieties showed increased relative contents of aldehydes, ketones, and esters and a variety of fruity aroma compounds. These compounds include hexanal, pentanal, nonanal, (E)-2-pentenal, 2-methyl-2-pentenal, octanal, benzaldehyde, (Z)-2-penten-1-ol, 1-hexanol, Linalool, geraniol, 1-octen-3-ol, phenylethyl alcohol, 3-penten-2-one, 2-nonanone, 3,5-octadien-2-one, 2-undecanone, ethyl hexanoate, methyl (E)-2-hexenoate, (E)-beta-damascenone, ethyl 2-hexenoate, (2-methoxyphenyl) butanoate, ethyl octanoate, methyl nonanoate, benzoic acid, methyl ester, hexanoic acid, (E)-3-hexenyl ester, α-Ocimene, and β-myrcene (Table 2). The molecular formulas, CAS numbers, and odor descriptions of these volatile compounds are presented in Table 2. The concentration and relative contents of these volatile compounds are presented in the given Appendix A.

A total of 154 aroma compounds were detected in the 16 different white tea varieties using GC×GC-O-MS. Among these, 133 compounds were successfully identified through the NIST library database, and 21 volatile compounds were listed as unknown (Table 2). The chromatographic results identified one nitrogen-containing compound, four heterocyclic compounds, 31 aldehydes, 37 alcohols, 19 ketones, five acids, 16 esters, 18 aromatic hydrocarbons and alkenes, two ethers, and 21 unknown compounds. The concentration of the unknown compounds was calculated based on an internal standard methodology, as mentioned in Section 2.6. The total relative concentrations of different classes of volatile organic compounds in the 16 different varieties of Yunnan white tea were calculated. For the OUC white tea variety, the total content of nitrogen-containing compounds was 0 µg/kg, the total content of heterocyclic compounds was 3136.03 µg/kg, the total aldehyde content was 19,094.29 µg/kg, the total alcohol content was 154,661.47 µg/kg, the total content of ketones was 7656.69 µg/kg, the total acid content was 2192.06 µg/kg, the total ester content was 20,831.14 µg/kg, the total content of aromatic hydrocarbons was 7512.57 µg/kg, the total content of ethers was 359.25 µg/kg, and the total content of unknown compounds was 2755.80 µg/kg. A comparative assessment of the contents of these compounds is presented in Figure 4A. The relative odor-active values (r-OAVs) of the identified volatile compounds are presented in Table 3.

### 3.4. Classification of Volatile Compounds in the 16 Yunnan High-Aroma White Teas

#### 3.4.1. Aldehydes

A total of 31 aldehydes were identified in the 16 white tea samples. Many aldehydes, such as hexanal, (Z)-3-hexenal, (E)-2-pentenal, and heptanal, contribute to the green aroma of white tea. Aldehydes like 2-methyl-2-pentenal, furfural, decanal, and neral also add delicate sweet aroma notes, which enhance the freshness and organoleptic properties of white tea. Aldehydes like pentanal and (E)-2-methyl-2-butenal had a significant nutty aroma, which plays an important role in delivering mellow flavor notes. The saturated aldehydes like decanal (sweet and floral), nonanal (waxy and rose-like), and heptanal (green, fresh, and herbal) with high odor threshold values contribute significantly to the overall aroma profile of white tea. It was found that the contents of aldehydes were significantly affected by the unshaken steam-cooked and compressed white tea varieties. Among these samples, other aldehydes increased slightly or even decreased depending on the processing condition.

#### 3.4.2. Alcohols

A large number of volatile compounds detected in all 16 white tea samples were classified as alcohols. A total of 37 alcohol compounds were identified in all white tea samples. The compounds specific to the alcohol group exhibited aroma notes such as ethereal, green, fresh, earthy, floral, and many others. Alcohols such as 2-heptanol, 3-hexen-1-ol, (Z)-3-hexen-1-ol, (E)-2-hexen-1-ol, cis-linaloloxide, (Z)-3-nonen-1-ol, and (E)-linalool oxide (pyranoid) exhibited delicate fresh aroma notes that contributed to the overall flavor profile of white tea samples. The results showed that the shaken samples had more aldehydes with notably higher relative concentrations. The relative odor-active values (r-OAVs) of most alcohols in the white tea samples were also significantly different between the shaken, unshaken, steam-cooked, and compressed white tea samples. For instance, 1-hexanol has a fruity and sweet aroma, and its r-OAV in OUC (compressed white tea) was 205, which was the highest among all the white tea samples. Similarly, the r-OAV of 1-octen-3-ol (mushroom-like and earthy aroma) was also highest (1704) in the OUC sample, while the lowest one (101) was in NSS7 (unshaken variety of white tea with small buds and leaves). Conversely, the highest r-OAV of linalool (448964) was reported in OUC, followed by OUS (188436) (green spring variety of loose white tea with big leaves). The lowest r-OAV of linalool was reported in NSS6 (60232). 1-octen-3-ol, linalool, geraniol, and phenylethyl alcohol, among other alcohols, had the highest r-OAVs, as presented in Table 3.

#### 3.4.3. Ketones

A total of 19 ketones were identified in the 16 different varieties of white tea from Yunnan province. Among these, 3-penten-2-one, 2-nonanone, 3,5-octadien-2-one, and 2-undecanone were classified as volatile compounds with fruity odors in white tea (Table 2). Compounds such as 2-decanone, α-ionone, and trans-β-ionone exhibited floral odor notes, while 3-octanone, 3-octen-2-one, (E)-3-octen-2-one, gamma-hexalactone, 2-butanone, and 4-(2,6,6-trimethyl-1-cyclohexen-1-yl) delivered a mix of odor notes, with herbal and earthy being the most common. The chromatography results revealed that the r-OAVs of 1-penten-3-one, (E)-beta-damascenone, and α-ionone were reportedly higher in the shaken varieties of white tea. (E)-beta-damascenone had the highest r-OAV of 63835 in OUS(the green spring variety of white tea with big leaves and was steam-cooked), whereas the lowest value of this compound (<1) was found in the SS1, SS4, SS5, SS7, NSS4, and NSS5 varieties. The highest r-OAV of α-ionone, which contributes to the sweet and floral aromas, was in OUC (247), as presented in Table 3.

#### 3.4.4. Heterocyclic Compounds

A total of four heterocyclic compounds were identified in the 16 different white tea varieties; these include 2-ethylfuran (beany and bready), 2-ethyl-5-methyl furan (burnt), 2-pentyl furan (beany and vegetable-like), and (E)-2-(2-pentenyl) furan (grassy and buttery). Among these, only the r-OAV of 2-pentyl furan was determined, with the highest value (431) observed in OUC and the lowest (7) one found in the NSS4 variety (Table 3). The results indicate that shaking, steam cooking, compressing, and other processing treatments used during the production of white tea can significantly influence the contents and odor-active values of heterocyclic compounds in white tea.

#### 3.4.5. Organic Acids

A total of five organic acids were identified in the 16 different white tea varieties, including acetic acid (pungent), propanoic acid (pungent and acidic), 3-methylbutyric acid (sour and sweet), hexanoic acid (sour and fatty), and pentanoic acid (acidic and tobacco-like). Among these, only the r-OAVs of 3-methylbutyric acid and hexanoic acid were determined. The highest r-OAV of hexanoic acid (194) was in SS6, and the highest r-OAV of 3-methylbutyric acid (5) was found in OUC (Table 3).

#### 3.4.6. Esters

A total of sixteen esters were identified in our study (Table 2), the majority of which exhibited fruity, fatty, and green aroma notes; these include ethyl hexanoate, methyl (E)-2-hexenoate, E-3-hexenyl acetate, ethyl 2-hexenoate, (2-methoxyphenyl) butanoate, ethyl octanoate, cis-3-hexenyl isovalerate, methyl nonanoate, benzoic acid-methyl ester, hexanoic acid-hexyl ester, (E)-hexanoic acid-3-hexenyl ester, and geranyl formate. Neryl acetate and methyl salicylate delivered a sweet odor note in white tea samples. Among these, the r-OAV values of methyl benzoate and methyl salicylate were determined. The highest r-OAV (759) among all esters belonged to methyl salicylate (sweet and aromatic) and was reported in the OUS variety (Table 3). The second-highest r-OAV (507) of methyl salicylate was reported in the OUC variety (Table 3). In addition, the r-OAV values of methyl benzoate in the shaken varieties were higher than in the unshaken samples (Table 3). Both these findings reveal that the shaking process increased the odor activities of esters in white tea varieties.

#### 3.4.7. Alkenes and Aromatic Hydrocarbons

A total of eighteen alkenes and aromatic hydrocarbons were identified in different white tea varieties. Among these, toluene (sweet), camphene (woody), ethylbenzene (floral), p-xylene (floral), β-myrcene (musty), α-phellandrene (herbal), limonene (citrusy), D-limonene (citrusy), β-phellandrene (terpenic and citrusy), trans-β-ocimene (herbal), γ-terpinene (woody), styrene (floral), o-cymene (citrusy and woody), terpinolene (woody), and p,α-dimethyl styrene (spicy and musty) were the most important compounds identified in the white tea samples. The highest r-OAV (2046) was reported for β-myrcene in OUC (the compressed variety), while the lowest (282) one was reported in NSS7. The highest r-OAV of ethylbenzene was 43 and reported in OUC, followed by SS7 (24), while the lowest values were observed in all other varieties. The highest r-OAV (38) of limonene was also found in OUC. The results revealed that the processing treatments can greatly alter the overall alkenes and aromatic hydrocarbons in the compared varieties.

#### 3.4.8. Hierarchical Clustering Analysis with Heat Maps

Figure 5A–C present a hierarchical clustering analysis prepared by plotting a heat map based on the relative concentrations of 154 volatile compounds detected in different white tea varieties. The data were normalized, and clustering was preprocessed using the Euclidean distance measure approach. Figure 5A presents the heat map analysis of the relative concentration of VOCs labeled as T1 to T50 in the 16 white tea varieties. Figure 5B depicts the contents of VOCs labeled as T51 to T100, and Figure 5C presents the quantities of VOCs labeled as T101 to T154. The heat map analysis demonstrates significant differences among the contents of volatile compounds across the 16 white tea samples. Compounds containing nitrogen and oxygen heterocyclic compounds fall within the range of T1–T5. Aldehydes are categorized in the range of T6 to T36, while alcohols range from T37 to T73. Ketones are classified as T74 to T92, and acids fall in the range of T93 to T97. Esters are labeled as T98 to T113, alkenes and aromatic hydrocarbons as T114 to T131, ethers as T132 to T133, and unknown compounds as T134 to T154. The detailed explanation of these terms are presented in Figure 5D. The OUC, NSS3, and SS6 white tea varieties showed high concentrations of the volatile compounds listed as T1 to T154. The results shown in the clustered analysis are also consistent with those of the OPLS-DA analysis (Figure 6A), highlighting the 154 (T1–T154) volatile compounds in the 16 different white tea varieties. The relative concentrations of the volatile compounds were rated on a scale ranging from red (1) to blue (0). The compounds ranging from T51 to T71 are categorized as alcohols, and their intensities (high and low) are presented in Figure 5B. The highest contents (red- and orange-colored cells) of these compounds were observed in OUC, and the lowest ones were found in SS6 and NSS4, NSS7, and SS6. The contents of certain alcohol compounds, such as T72 (nerolidol) and T73 (T-muurolol), along with ketones like T75 (3-penten-2-one), T76 (2,5-dimethyl-3-hexanone), and T77 (3-octanone), were reportedly low in the OUC white tea variety. The compounds ranging from T114 (toluene) to T121 (α-Terpinene) belong to the aromatic hydrocarbon and alkene group, and the highest concentration of these compounds was observed in the OUC white tea variety, as visualized by the red-colored cells (Figure 5C). The compounds ranging from T134 to T154 represent the heat map distribution of unknown compounds across the 16 white tea verities; their relative contents are presented in Figure 5C.

### 3.5. OPLS-DA Analysis of Different White Tea Varieties

The volatile organic profiles of different white tea varieties identified through GC×GC-O-MS were further analyzed for the variation in the datasets and to find differences among the samples through orthogonal partial least squares discriminant analysis (OPLS-DA). Therefore, to investigate the effect of the shaking treatment on white tea varieties and classify the 16 different varieties, a systematic pattern recognition method was used to differentiate between the datasets. Figure 6A presents the score plots of the first principal component and presents the variance in white tea chromatographic datasets. The aroma profiles of white tea varieties were differentiated using their variable importance projection (VIP) values. The prediction factors (R^2^Y = 0.670) and the prediction goodness (Q^2^ = 0.648) of the applied OPLS-DA model indicate its validity and stability and suggest that it can be used to further screen out the differential samples. The shaken samples SS6 and NSS4 are found in the uppermost section of the score plot. The shaken samples SS2 and SS7 and the unshaken samples NSS7 and NSS6 are located in the bottom left section of the score plot, while the shaken samples SS1, SS3, SS4, and SS5 are located in the middle section of the abscissa of the loading plots. The samples OUS and OUC are located in the upper and bottom right sections of the OPLS-DA score plot (Figure 6A). A distinct separation between the shaken and unshaken varieties of white tea is shown in the score plots. A UV-scaling model was selected for this study. The cumulative variations described in the Y-matrix and Z-matrix were 0.670 (R^2^ Y_cum_) and 0.926 (R^2^ X_cum_), respectively. The model’s cross-validated predictive ability (Q^2^_cum_) was 0.648. Furthermore, the values of Q^2^ R^2^ Y_cum_ were greater than 0, which shows that the OPLS-DA model was valid for the differentiation of different white tea varieties. The score plot indicates that the shaking treatment significantly altered the volatile organic profile of white tea varieties. Additionally, the cluster analysis showed remarkable differences among different white tea varieties and confirms the application of the shaking treatment in improving the aroma profile of white tea.

## 4. Discussion

Tea’s aroma and quality are influenced by its geographical origin and the production and processing methods. The sensory results from the current study were compared to those of the existing literature, and several key similarities emerged, particularly regarding the role of shaking in the sensory characteristics of tea. A recent study also found that shaking can significantly improve the flavor quality of black tea. The shaken black tea varieties exhibited a fruity and flowery aroma, which was considered superior to the traditional white tea variety, which is characterized by a sweet aroma [33]. Another study analyzed the impact of the shaking and standing treatments on the aroma characteristics of summer black tea. The volatile metabolite results indicate that shaking promotes the accumulation of volatile organic compounds and improves the floral and sweet odors of summer black tea [34]. The content of volatiles in processed tea varies with the processing methods applied [35]. The quality of white tea is determined by its organoleptic and sensory properties. The white tea varieties analyzed through sensory evaluation in our study exhibited strong floral, herbal, and woody aromas, with OUC and OUS having the highest scores. E-nose analysis identified significant differences in aromatic alcoholic VOCs between the unshaken and shaken white tea varieties. Previous researchers identified nine key odor-active compounds in white tea, including 2-methyl-butanal, dimethyl sulfide, 1-penten-3-one, (Z)-4-heptenal, hexanal, β-ionone, linalool, β-myrcene, and geraniol. These compounds help distinguish white tea produced by different withering methods. Their key aroma findings were in accordance with the current study [1]. OAV calculations and aroma recombinant experiments from a previous study identified 15 potent aroma-active compounds responsible for the distinct aroma of four white tea varieties with fruity, floral, sweet woody, and fermented odor characteristics. Linalool, geraniol, and amino acid reaction products were the key contributors to these distinct aromas in their study [13].

It is well-known that ketones and alcohols contribute significantly to the floral aroma of teas. Oxidation also affects the flavor profile of white tea [16,36]. Our study identified 37 alcohol volatile compounds in white tea varieties, with shaking treatments significantly impacting their aroma profiles and contributing to the tea’s flavor. The present study also identified 19 different ketones in the 16 white tea varieties. It was believed that these ketones, which form during oxidation, contribute greatly to the white tea’s diverse flavor profile. Many ketones are formed during the oxidation process of white tea; this may occur due to air exposure during processing or even storage [16,27]. Ketone volatiles significantly affect the overall aroma and flavor profile of white tea. Because of their delicate fruity, herbal, and flowery aroma profiles, ketones contribute to unique fragrant notes in white tea [3]. The odor thresholds (OTs) of potent ketones, as discussed in the literature, include α-ionone (flowery and violet-like), (E)-beta-damascenone (apple- and honey-like), and 1-penten-3-one (garlic-like), with thresholds of 0.00378 mg/kg, 0.000002 mg/kg, 0.1 mg/kg, and 0.023 mg/kg respectively [37].

Although white tea is the least processed type of tea, the contents and types of ketones in white teas are directly influenced by some important factors, such as the cultivar, fermentation level, and processing methods. Esters exhibit a coconut-like and sweet odor, while aldehydes contribute to the green and citrusy odor notes [38]. It was also predicted that aldehydes in white tea samples contribute to a dual behavior, i.e., these aldehydes not only contribute to the odor profile individually but also actively interact with other aroma compounds to generate an appealing and pleasant aroma profile in white tea during processing [20]. Shaking could also promote the release of volatile compounds by exhibiting fruity and floral aroma notes and can ultimately change the overall content of these volatile compounds by enhancing the organoleptic properties of white tea [1]. The role of the shaking treatment on the aroma quality of alcohol in black tea has been explored previously. The proportional contents of alcohol in the shaken varieties of white teas were higher (3113.19 ± 339.82 μg/L) than those of the black tea varieties that were processed with traditional methods (3491.52 ±163.79 μg/L) [39]. A study investigating the potential of the shaking treatment on the flavor and fragrance profile of oolong tea revealed that the shaking treatment affected the overall contents and that the chemical transformation of the alcoholic compounds helped enhance the characteristic aroma of oolong tea [3].

Heterocyclic volatile compounds are easily detected through the human nose and ODP because of their intense aroma. These compounds are produced in white tea by either amino acid-derived pathways or by Maillard reactions [40,41]. Our study revealed that the shaking treatment significantly influences the contents of heterocyclic compounds and odor-active values in white tea varieties. Furans are an important heterocyclic class of volatile compounds found in white tea. Furan derivatives were the most common heterocyclic compounds (7.939%) to be reported in white tea samples [21]. Organic acids such as short-chain fatty acids (both straight and branched chains) are an important class of volatile compounds but are found in low quantities in white tea. These compounds are well-known for delivering a sour and acidic taste to white tea. Hexanoic acid is the key organic acid found in different white tea varieties. As the duration of the fermentation of white tea increases, it becomes more acidic and less sweet in taste [42].

It is well-known that the oxidation process has notable effects on the acidic profile of teas. The shaking process can directly increase the expression of oxidase, which increases oxidation [3]. The contents of organic acids in fresh white tea leaves were reportedly much higher than in processed white tea. The current study revealed that the shaking process increased the odor activities of 16 identified esters, delivering diverse aroma notes in the shaken white tea varieties. Various types of esters are found in white tea, including esters of acetic acid and butyric acid as well as phthalate esters (PAEs). PAEs are largely found in the mature and aged leaves of tea plants [43]. It is also believed that esters accumulate in white tea during its withering process [6]. In our study, 18 aromatic hydrocarbons were identified in white tea samples, revealing that processing treatments significantly alter the overall contents of alkenes. Polycyclic aromatic hydrocarbons (PAHs) are organic compounds found in the leaves of white tea, and almost 1.6% of the hydrocarbons in the teas are released during the tea infusion process [44]. Trace levels of alkenes are naturally found in the tea leaves because they are part of plant lipids. However, processing treatments, including withering, shaking, and oxidation, might alter the contents of aromatic alkenes in white tea. Research has shown that white tea has a higher PAH content (24–119 μg/kg) compared to green tea (3.1–92 μg/kg) but a lower PAH content compared to mate tea (194–1795 μg/kg) and black tea (1.8–186 μg/kg) [45].

OPLS-DA is a promising multivariate chemometric approach for analyzing complex chromatography datasets and identifying the key factors that can be used for the classification of these datasets. OPLS-DA is ideal for the classification of datasets that have linear or multi-collinear variables, such as gas chromatography–mass spectrometry fingerprint datasets [20,31,46]. The OPLS-DA model used in this study successfully differentiated the aroma profiles of shaken and unshaken white tea varieties. The model revealed that the shaking treatment significantly affected the volatile organic profiles of white tea samples. The key outcomes indicated that the model can successfully explain 67% of the variation in the dependent variable and predict approximately 64.8% of unseen data. Previously, an OPLS-DA model was applied to investigate the changes in the volatile profile of fresh-scent green tea. The OPLS-DA parameters showed strong model validity and capability (R^2^Y = 0.935 and Q^2^Y = 0.759) [47].

## 5. Conclusions

The present study investigated special processed varieties of white tea using two-dimensional GC×GC-O-MS, e-nose, and chemometric analyses. The white tea varieties obtained from the Lüchun county of Yunnan province of Mainland China were grouped into three major categories (shaken, unshaken, and other processed varieties). The human descriptive sensory and e-nose analyses revealed that the shaking treatment has a remarkable impact on the aroma profile of different white tea varieties. Interestingly, it was also revealed that woody, floral, and herbal odors are mainly considered potent aroma descriptors of white tea. The e-nose analysis revealed that the highest signals were captured on the W5S and WS2 sensors, representing sensitivity towards aromatic and alcoholic VOCs. The chromatographic results revealed a total of 154 volatile compounds in 16 different varieties of white tea using GC×GC-O-MS. The concentration of individual volatile compounds was calculated; the relative odor-active values (r-OAVs) of the identified volatile compounds were also calculated but were only presented if they were greater than 1 (>1). The key odorants in each white tea variety were highlighted. Additionally, orthogonal partial least squares discriminant analysis (OPLS-DA) was applied to identify the differences in the content of volatile compounds between shaken and unshaken white tea varieties. Future mechanistic studies will focus on investigating the formation pathways and precursors responsible for generating key odor-active compounds in shaken white tea varieties.

## Figures and Tables

**Figure 1 foods-14-00271-f001:**
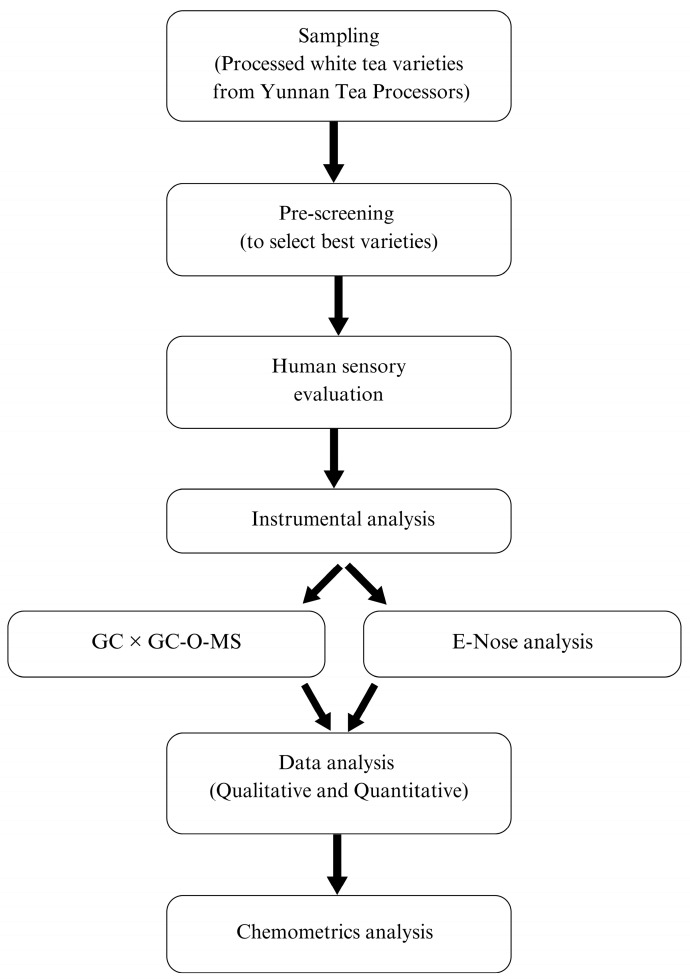
Sequential flowchart of white tea aroma and sensory characterization.

**Figure 2 foods-14-00271-f002:**
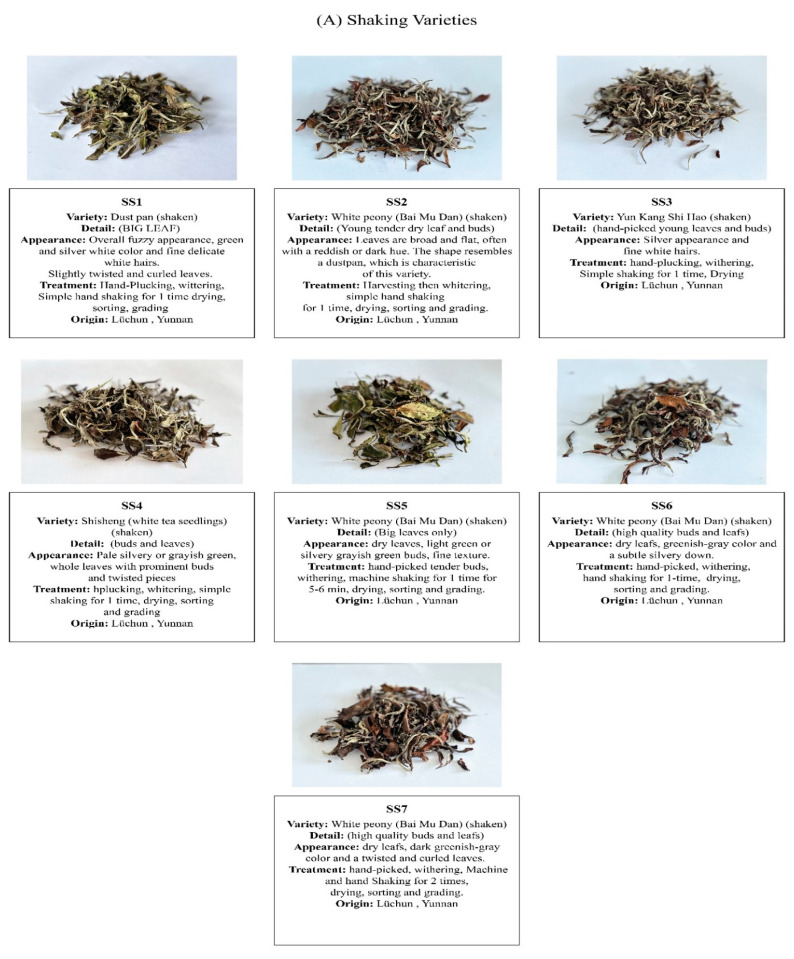
(**A**) Detailed information about seven shaken commercial Yunnan white tea varieties. (**B**) Detailed information about seven unshaken commercial Yunnan white tea varieties. (**C**) Detailed information about two other commercial Yunnan white tea varieties.

**Figure 3 foods-14-00271-f003:**
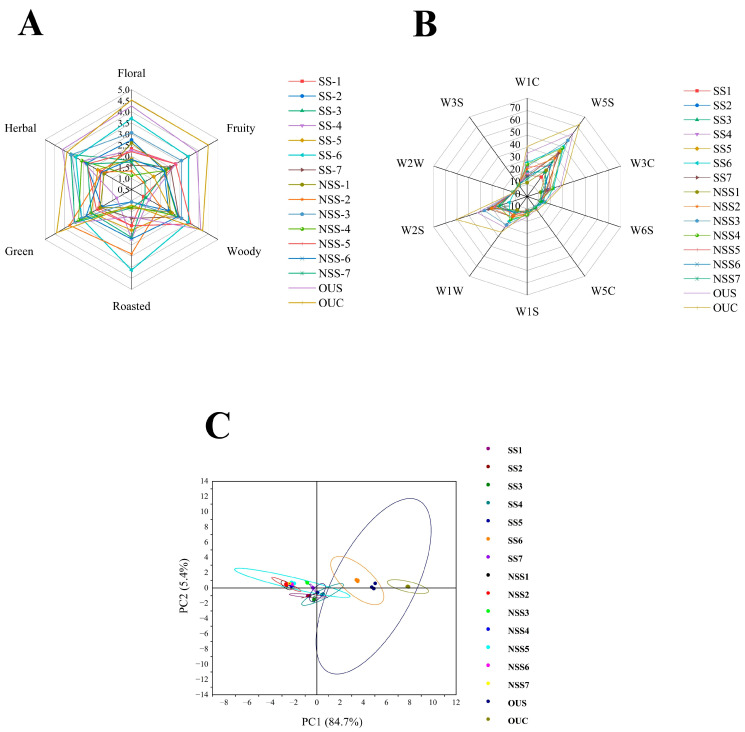
(**A**) Sensory evaluation scores of the 16 white tea varieties. (**B**) E-nose radar map of the white tea samples. (**C**) PCA of the e-nose data.

**Figure 4 foods-14-00271-f004:**
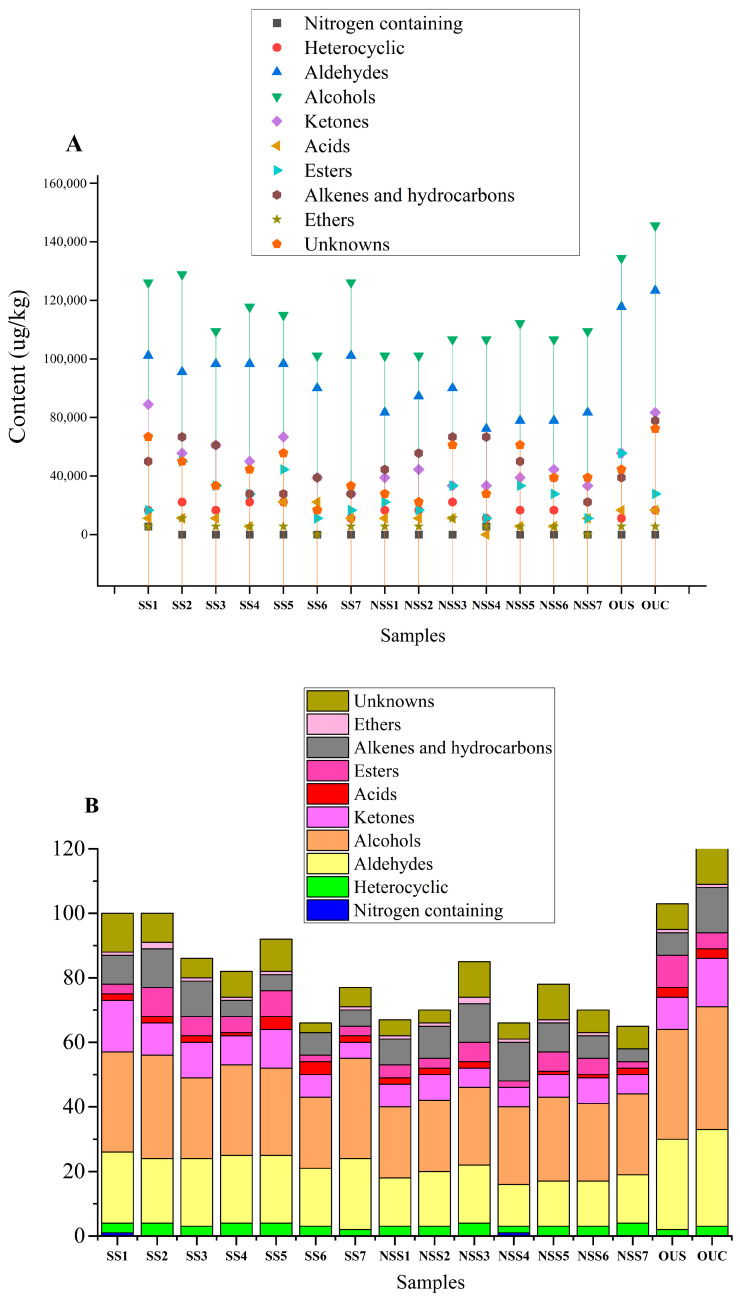
(**A**) Relative contents of VOCs in the 16 white tea varieties. (**B**) Classification of VOCs in the 16 different white tea varieties.

**Figure 5 foods-14-00271-f005:**
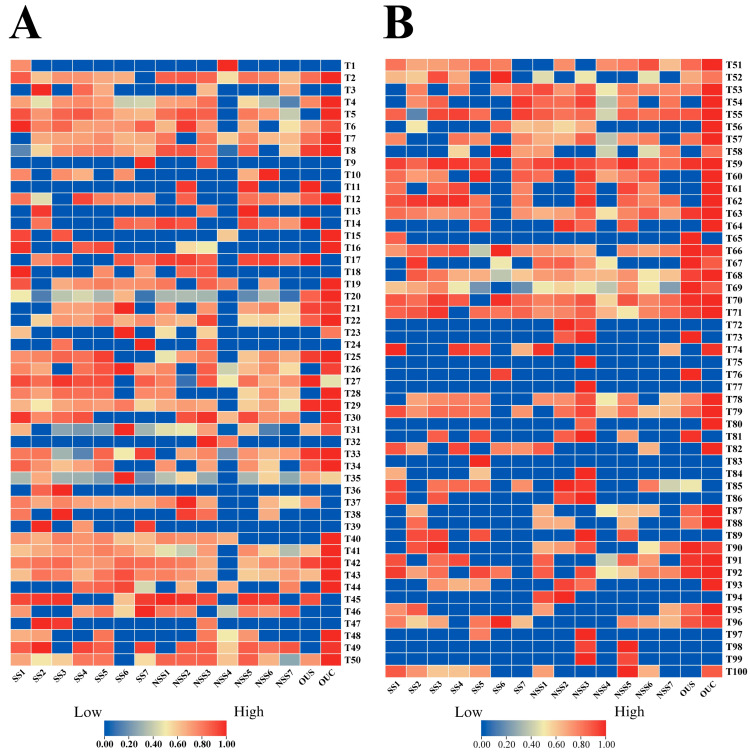
(**A**). Clustered heat map analysis of volatiles labeled as T1 to T50. (**B**) Clustered heat map analysis of volatiles labeled as T51 to T100. (**C**) Clustered heat map analysis of volatiles labeled as T101 to T154. (**D**) Decoded terms for the VOC classes of T1 to T154.

**Figure 6 foods-14-00271-f006:**
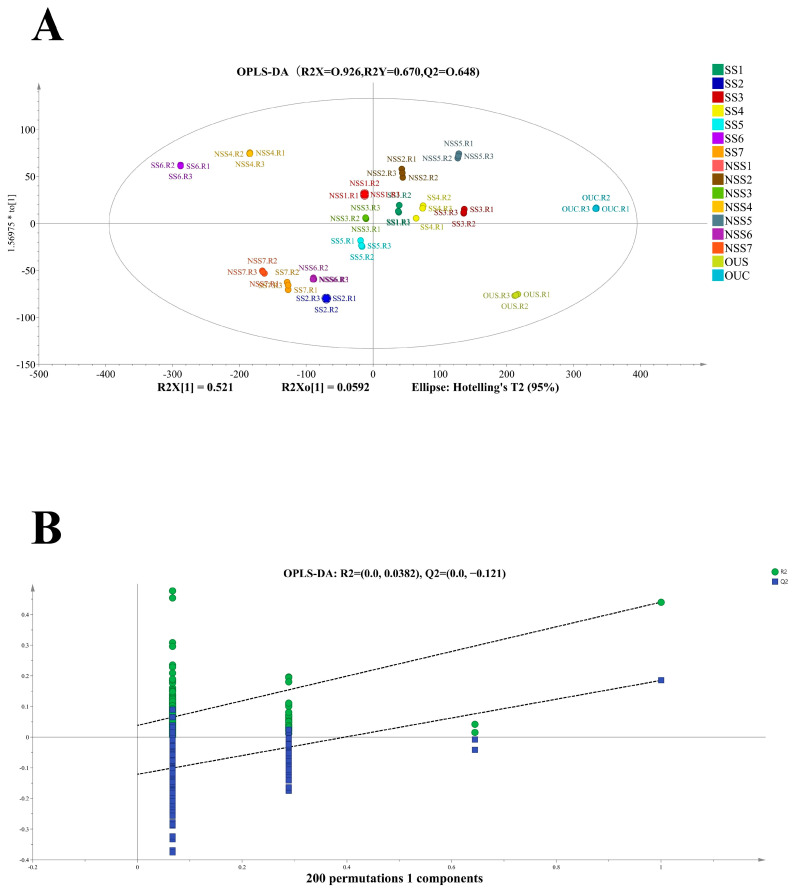
(**A**) OPLS-DA analysis of 16 different white tea varieties. (**B**) The permutations plot of OPLS-DA (green dots represent R2 and indicate the predicted classes, while blue dots represent Q2 and indicate the actual training classes).

**Table 1 foods-14-00271-t001:** Detailed information about odor attributes used in the sensory training for white tea samples with reference standard compounds.

Descriptor	Definition	Reference Compound
Woody	A smell similar to sawdust odor or peanut shells	(E)-geranyl acetone CAS No: 3796-70-1
Floral	A smell affiliated with the odor of jasmine and roses	Phenyl acetaldehyde CAS No: 122-78-1
Fruity	A smell associated with the odor of fresh fruit and jujube	Geraniol CAS No: 106-24-1
Nutty	A smell associated with almonds and nut-like notes	2 Ethyl 3,5-6 dimethyl pyrazine CAS No: 13925-07-0
Grassy/fresh leaves	A smell associated with freshly cut grass	Hexanal CAS No: 66-25-1
Herbaceous	A light, fresh, and aromatic scent that is associated with traditional herbs	Natural mugwort

**Table 2 foods-14-00271-t002:** List of aroma compounds identified in the 16 different varieties of white tea.

No.	RT	LRI-Calculated	LRI-NIST	Δ LRI	Compounds	CAS	Odor	Formula
**Nitrogen-containing compounds**								
T1	9.4837	1132	1162	30	1-Ethyl pyrrole	617-92-5	Burnt and smoky	C_6_H_9_N
**Oxygen heterocyclic compounds**								
T2	4.4841	920	945	25	2-Ethylfuran	3208-16-0	Beany and cocoa-like	C_6_H_8_O
T3	5.8178	1003	1024	21	2-Ethyl-5-methyl furan	1703-52-2	Fresh and burnt	C_7_H_10_O
T4	11.1341	1184	1213	29	2-Pentyl furan	3777-69-3	Beany and vegetable-like	C_9_H_14_O
T5	13.8175	1257	1282	25	(E)-2-(2-pentenyl) furan	70424-14-5	Grassy and buttery	C_9_H_12_O
**Aldehydes**								
T6	4.1508	910	918	8	3-Methylbutanal	590-86-3	Aldehydic and peachy	C_5_H_10_O
T7	4.9016	947	953	6	Pentanal	110-62-3	Bready and nutty	C_5_H_10_O
T8	6.8676	1040	1048	8	Hexanal	66-25-1	Grassy and fatty	C_6_H_12_O
T9	7.1625	1049	1069	20	(E)-2-methyl-2-butenal	497-03-0	Green and nutty	C_5_H_8_O
T10	7.4009	1062	1088	26	2-Methyl-2-butenal	1115-11-3	Musty and vegetable-like	C_5_H_8_O
T11	8.1505	1090	1114	24	(Z)-3-hexenal	6789-80-6	Green and leafy	C_6_H_10_O
T12	8.1805	1090	1096	6	(E)-2-pentenal	1576-87-0	Green and fruity	C_5_H_8_O
T13	8.951	1117	1142	25	2-methyl-2-pentenal	623-36-9	Sweet and fruity	C_6_H_10_O
T14	9.5679	1138	1163	25	Heptanal	111-71-7	Green and herbal	C_7_H_14_O
T15	9.7341	1140	1150	10	5-Methyl hexanal	1860-39-5	Warm spicy	C_7_H_14_O
T16	10.8177	1173	1188	15	(E)-2-hexenal	6728-26-3	Apple-like and green	C_6_H_10_O
T17	10.8324	1173	1193	20	2-Hexenal	505-57-7	Sweet and almond-like	C_6_H_10_O
T18	11.6508	1199	1216	17	(Z)-4-heptenal	6728-31-0	Fatty and green	C_7_H_12_O
T19	13.3672	1247	1263	16	Octanal	124-13-0	Fatty and fruity	C_8_H_16_O
T20	17.5676	1354	1365	11	Nonanal	124-19-6	Waxy and rose-like	C_9_H_18_O
T21	17.5676	1357	1367	10	(E,E)-2,4-hexadienal	142-83-6	Sweet and spicy	C_6_H_8_O
T22	18.7347	1397	1412	15	(E)-2-octenal	2548-87-0	Fresh and spicy	C_8_H_14_O
T23	19.1505	1409	1420	11	α-Cyclocitral	432-24-6	Mushroom-like	C_8_H_16_O
T24	19.9841	1413	1422	9	Furfural	98-01-1	Sweet and woody	C_5_H_4_O_2_
T25	21.1511	1444	1454	10	(E,E)-2,4-heptadienal	4313--03--5	Green and vegetable-like	C_7_H_10_O
T26	21.6177	1465	1471	6	Decanal	112-31-2	Sweet and floral	C_10_H_20_O
T27	22.5673	1467	1478	11	Benzaldehyde	100-52-7	Fruity	C_7_H_6_O
T28	23.0673	1488	1509	21	(E)-2-nonenal	18829-56-6	Fatty and green	C_9_H_16_O
T29	24.9005	1535	1551	16	(E,Z)-2,6-nonadienal	557-48-2	Cucumber-like and violet-like	C_9_H_14_O
T30	26.0344	1565	1590	25	Beta-cyclocitral	432-25-7	Rose-like and saffron-like	C_10_H_16_O
T31	26.9677	1588	1616	28	Safranal	116-26-7	Fresh and herbal	C_10_H_14_O
T32	27.234	1595	1597	2	(E)-2-decenal	3913-81-3	Waxy and earthy	C_10_H_18_O
T33	27.4845	1599	1605	6	Benzeneacetaldehyde	122-78-1	Green and floral	C_8_H_8_O
T34	28.2173	1624	1653	29	2-Butyl-2-octenal	13019-16-4	Aldehydic and green	C_12_H_22_O
T35	30.2839	1652	1660	8	Neral	106-26-3	Sweet and citrusy	C_10_H_16_O
T36	32.9849	1749	1767	18	(E,E)-2,4-decadienal	2363-88-4	Orange-like	C_10_H_16_O
**Alcohols and heterocylic compounds**								
T37	9.3675	1127	1129	2	1-Penten-3-ol	616-25-1	Green and buttery	C_5_H_10_O
T38	10.5674	1173	1184	11	3-Methyl-1-butanol	123-51-3	Whiskey-like and malt-like	C_5_H_12_O
T39	10.9012	1176	1191	15	2-Methyl-1-butanol	137-32-6	Whiskey-like and malt-like	C_5_H_12_O
T40	12.3673	1216	1217	1	1-Pentanol	71-41-0	Balsamic	C_5_H_12_O
T41	14.7508	1279	1286	7	2-Heptanol	543-49-7	Fresh and lemongrass-like	C_7_H_16_O
T42	14.8841	1280	1304	24	(Z)-2-penten-1-ol	1576-95-0	Fruity and cherry-like	C_5_H_10_O
T43	15.9174	1313	1316	3	1-Hexanol	111-27-3	Ethereal fruity	C_6_H_14_O
T44	16.984	1343	1372	29	3-Hexen-1-ol	0544-12-7	Fresh and grassy	C_6_H_12_O
T45	17.0844	1356	1351	5	(Z)-3-hexen-1-ol	928-96-1	Fresh and grassy	C_6_H_12_O
T46	17.8173	1367	1380	13	(E)-2-hexen-1-ol	928-95-0	Fresh and leafy	C_6_H_12_O
T47	18.2348	1375	1390	15	(Z)-2-hexen-1-ol	928-94-9	Green and beany	C_6_H_12_O
T48	18.5676	1384	1412	28	2-Octanol	123-96-6	Spicy and green	C_8_H_18_O
T49	19.4008	1406	1425	19	(Z)-linalool oxide (furanoid)	5989-33-3	Earthy and floral	C_10_H_18_O_2_
T50	19.6505	1415	1428	13	1-Octen-3-ol	3391-86-4	Mushroom-like and earthy	C_8_H_16_O
T51	19.9008	1417	1447	30	1-Heptanol	111-70-6	Musty and leafy	C_7_H_16_O
T52	20.1302	1423	1465	42	cis-Linaloloxide	N/A	Floral	C_10_H_18_O_2_
T53	20.1346	1427	1454	27	6-Methylhept-5-en-2-ol	1569-60-4	Sweet and oily	C_8_H_16_O
T54	20.3176	1431	1460	29	Nerol oxide	1786-08-9	Green and grassy	C_10_H_16_O
T55	20.5674	1433	1440	7	(E)-linalool oxide (furanoid)	34995-77-2	Floral	C_10_H_18_O_2_
T56	21.2346	1459	1470	11	2-Ethyl-1-hexanol	104-76-7	Citrusy and fresh	C_8_H_18_O
T57	22.5673	1478	1484	6	2-Nonanol	628-99-9	Waxy and creamy	C_9_H_20_O
T58	23.1337	1491	1496	5	Dihydrolinalool	18479-51-1	Citrusy and floral	C_10_H_18_O2
T59	23.7339	1505	1519	14	Linalool	78-70-6	Citrusy and floral	C_10_H_18_O_2_
T60	23.9842	1512	1519	7	1-Octanol	111-87-5	Waxy and orange-like	C_8_H_18_O
T61	25.5671	1552	1572	20	4-Terpinenol	562-74-3	Peppery and woody	C_10_H_18_O_2_
T62	25.7338	1556	1585	29	1,5,7-Octatrien-3-ol, 3,7-dimethyl-	29957-43-5	Earthy	C_10_H_16_O
T63	27.8173	1627	1619	8	1-Nonanol	0143-08-08	Rose-like and orange-like	C_9_H_20_O
T64	29.0674	1643	1654	11	α-Terpineol	98-55-5	Pine-like and terpenic	C_10_H_18_O_2_
T65	29.4839	1655	1663	8	2,6-Octadien-1-ol, 3,7-dimethyl-, formate, (2Z)-	2142-94-1	Citrusy and floral	C_11_H_18_O2
T66	31.4008	1711	1719	8	(E)-linalool oxide (pyranoid)	39028-58-5	Woody and fresh	C_10_H_18_O_2_
T67	31.6509	1712	1735	23	Citronellol	106-22-9	Rose-like and floral	C_10_H_20_O
T68	32.7205	1740	1767	27	Nerol	106-25-2	Sweet and neroli-like	C_10_H_18_O_2_
T69	34.4006	1785	1797	12	Geraniol	106-24-1	Floral and fruity	C_10_H_18_O_2_
T70	34.9005	1801	1828	27	Benzyl alcohol	100-51-6	Floral and sweet	C_7_H^8^O
T71	36.1508	1834	1860	26	Phenylethyl Alcohol	60-12-8	Floral and rose-like	C_8_H_10_O
T72	40.9838	2006	2007	1	Nerolidol	7212-44-4	Floral and woody	C_15_H_26_O
T73	45.4005	2156	2160	4	T-muurolol	19912-62-0	Herbal and spicy	C_15_H_26_O
**Ketones**								
T74	5.5677	988	1018	30	1-Penten-3-one	1629-58-9	Peppery and garlicky	C_5_H_8_O
T75	8.1502	1090	1109	19	3-Penten-2-one	625-33-2	Fruity	C_5_H_8_O
T76	9.6661	1138	1145	7	2,5-Dimethyl-3-hexanone	1888-57-9	Orange-like and fresh	C_8_H_16_O
T77	12.2336	1215	1230	15	3-Octanone	106-68-3	Herbal	C_8_H_16_O
T78	14.4842	1273	1282	9	Cistus cyclohexanone	2408-37-9	Labdanum-like and honey-like	C_9_H_16_O
T79	15.1344	1293	1313	20	6-Methyl-5-hepten-2-one	110-93-0	Citrusy and lemony	C_8_H_14_O
T80	17.4006	1357	1379	22	2-Nonanone	821-55-6	Fruity and herbal	C_9_H_18_O
T81	17.9009	1363	1363	0	3-Octen-2-one	1669-44-9	Earthy and spicy	C_8_H_14_O
T82	17.9676	1366	1363	3	(E)-3-octen-2-one	18402-82-9	Earthy and mushroom-like	C_8_H_14_O
T83	18.9836	1396	1373	23	5-Decanone	820-29-1	Creamy	C_10_H_20_O
T84	21.5674	1456	1482	26	2-Decanone	693-54-9	Floral	C_10_H_20_O
T85	24.2339	1518	1516	2	3,5-Octadien-2-one	38284-27-4	Fruity	C_8_H_12_O
T86	25.7338	1557	1578	21	2-Undecanone	112-12-9	Waxy and fruity	C_11_H_22_O
T87	26.9838	1588	1607	19	Acetophenone	98-86-2	Almond-like	C_8_H_8_O
T88	28.7341	1634	1664	30	Gamma-hexalactone	0695-06-07	Herbal and coconut-like	C_6_H_10_O_2_
T89	29.0339	1648	1660	12	Tetrahydrogeranyl acetone	1604-34-8	Dry and musty	C_13_H_26_O
T90	33.4012	1760	1788	28	(E)-beta-damascenone	23726-93-4	Apple-like and rose-like	C_13_H_18_O
T91	34.5672	1792	1819	27	α-Ionone	127-41-3	Sweet and woody	C_13_H_20_O
T92	37.4841	1875	1897	22	trans-β-Ionone	79-77-6	Dry powdery	C_13_H_20_O
**Acids**								
T93	19.2171	1404	1415	11	Acetic acid	64-19-7	Sharp and pungent	C_2_H_4_O_2_
T94	22.4671	1480	1493	13	Propanoic acid	79-09-4	Pungent and acidic	C3H6O2
T95	27.6507	1605	1633	28	3-Methylbutyric acid	503-74-2	Sour and sweet	C_5_H_10_O_2_
T96	33.9843	1776	1800	24	Hexanoic acid	142-62-1	Sour and fatty	C_6_H_12_O_2_
T97	33.9841	1776	1768	8	Pentanoic acid	109-52-4	Acidic and sharp	C_5_H_10_O_2_
**Esters**								
T98	11.4548	1194	1212	18	Ethyl hexanoate	123-66-0	Fruity and banana-like	C_8_H_16_O_2_
T99	13.4839	1247	1251	4	Methyl 2-hexenoate	2396-77-2	Fatty and earthy	C_8_H_16_O_2_
T100	13.4842	1247	1272	25	Methyl (E)-2-hexenoate	13894-63-8	Fatty and fruity	C_7_H_12_O_2_
T101	14.6174	1277	1300	23	E-3-hexenyl acetate	3681-82-1	Fruity and green	C_8_H_14_O_2_
T102	15.5507	1305	1328	23	Ethyl 2-hexenoate	1552-67-6	Fatty and fruity	C_8_H_14_O_2_
T103	18.8177	1401	1415	14	(2-methoxyphenyl) butanoate	10032-15-2	Green and fruity	C_11_H_22_O_2_
T104	19.2005	1406	1412	6	Ethyl octanoate	106-32-1	Fruity and banana-like	C_10_H_20_O_2_
T105	21.2346	1448	1440	8	Cis-3-Hexenyl isovalerate	35154-45-1	Fresh and apple-like	C_11_H_20_O_2_
T106	21.6507	1458	1486	28	Methyl nonanoate	1731-84-6	Fruity	C_10_H_20_O_2_
T107	25.9009	1563	1589	26	Methyl benzoate	93-58-3	Fruity and cherry-like	C_8_H_8_O_2_
T108	26.3841	1576	1596	20	Hexyl hexanoate	6378-65-0	Herbal and fresh	C_12_H_24_O_2_
T109	28.5674	1630	1645	15	trans-2-Hexenyl hexanoate	53398-86-0	Green and fruity	C_12_H_22_O_2_
T110	29.3836	1655	1681	26	Geranyl formate	105-86-2	Fresh and rose-like	C_11_H_18_O_2_
T111	31.3176	1706	1714	8	Neryl acetate	0141-12-8	Sweet and citrusy	C_12_H_20_O_2_
T112	31.6511	1711	1735	24	Methyl salicylate	119-36-8	Sweet and aromatic	C_8_H_8_O_3_
T113	48.2344	2208	2220	12	Ethyl hexadecanoate	628-97-7	N/A	C_18_H_36_O_2_
**Alkenesand aromatic hydrocarbons**								
T114	5.9845	1009	1017	8	Toluene	108-88-3	Sweet	C_7_H_8_
T115	6.6511	1034	1040	6	Camphene	79-92-5	Woody and terpenic	C_10_H_16_
T116	7.7955	1077	1098	21	Ethylbenzene	100-41-4	Aromatic and floral	C_8_H_10_
T117	8.2342	1090	1119	29	p-Xylene	106-42-3	Sweet and floral	C_8_H_10_
T118	8.6838	1118	1136	19	β-Myrcene	123-35-3	Balsamic and spicy	C_10_H_16_
T119	8.7675	1120	1150	30	α-Phellandrene	99-83-2	Terpenic and citrusy	C_10_H_16_
T120	9.5241	1136	1164	28	Limonene	138-86-3	Citrusy and orange-like	C_10_H_16_
T121	9.5674	1138	1130	8	α-Terpinene	99-86-5	Woody and lemony	C_10_H_16_
T122	10.1507	1154	1175	21	D-Limonene	5989-27-5	Citrusy and orange-like	C_10_H_16_
T123	10.6158	1168	1172	4	β-Phellandrene	0555-10-2	Terpenic and citrusy	C_10_H_16_
T124	11.4006	1194	1222	28	trans-β-Ocimene	3779-61-1	Sweet and herbal	C_10_H_16_
T125	11.8176	1204	1213	9	γ-Terpinene	99-85-4	Oily and woody	C_10_H_16_
T126	12.0345	1210	1236	26	Styrene	100-42-5	Sweet and floral	C_8_H_8_
T127	12.3516	1213	1237	24	α-Ocimene	502-99-8	Fruity and floral	C_10_H_16_
T128	12.6509	1226	1254	28	o-Cymene	527-84-4	Citrusy and spicy	C_10_H_14_
T129	12.9005	1241	1258	17	Terpinolene	586-62-9	Woody and floral	C_10_H_16_
T130	17.2348	1348	1366	18	(E,E)-2,6-Alloocimene	3016-19-1	Terpenic and sweet	C_10_H_16_
T131	18.8177	1399	1417	18	p,α-dimethyl styrene	1195-32-0	Spicy and musty	C_10_H_12_
**Ethers**								
T132	23.3672	1496	1496	0	Theaspirane	36431-72-8	Tea and herbal	C_13_H_22_O
T133	27.9837	1615	1624	9	Estragole	140-67-0	Sweet and spicy	C_10_H_12_O
**Unknown compounds**								
T134	25.9837	1565	N/A	N/A	Unknown-1	N/A	N/A	N/A
T135	26.2344	1572	N/A	N/A	Unknown-2	N/A	N/A	N/A
T136	30.3171	1674	N/A	N/A	Unknown-3	N/A	N/A	N/A
T137	25.9841	1563	N/A	N/A	Unknown-4	N/A	N/A	N/A
T138	25.9846	1563	N/A	N/A	Unknown-5	N/A	N/A	N/A
T139	28.5506	1632	N/A	N/A	Unknown-6	N/A	N/A	N/A
T140	29.651	1659	N/A	N/A	Unknown-7	N/A	N/A	N/A
T141	29.7342	1661	N/A	N/A	Unknown-8	N/A	N/A	N/A
T142	30.9013	1692	N/A	N/A	Unknown-9	N/A	N/A	N/A
T143	32.2341	1730	N/A	N/A	Unknown-10	N/A	N/A	N/A
T144	11.5673	1197	N/A	N/A	Unknown-11	N/A	N/A	N/A
T145	14.5338	1287	N/A	N/A	Unknown-12	N/A	N/A	N/A
T146	22.0151	1467	N/A	N/A	Unknown-13	N/A	N/A	N/A
T147	24.5509	1520	N/A	N/A	Unknown-14	N/A	N/A	N/A
T148	28.5676	1624	N/A	N/A	Unknown-15	N/A	N/A	N/A
T149	33.9843	1776	N/A	N/A	Unknown-16	N/A	N/A	N/A
T150	37.9837	1890	N/A	N/A	Unknown-17	N/A	N/A	N/A
T151	14.9009	1284	N/A	N/A	Unknown-18	N/A	N/A	N/A
T152	29.1506	1646	N/A	N/A	Unknown-19	N/A	N/A	N/A
T153	7.3169	1059	N/A	N/A	Unknown-20	N/A	N/A	N/A
T154	39.1511	1924	N/A	N/A	Unknown-21	N/A	N/A	N/A

RT: Retention time. LRI-Calculated: The calculated linear retention indices of volatile compounds. LRI-NIST: The compounds LRI from the NIST database. CAS: Chemical Abstracts Service number. Δ-LRI: Difference between calculated LRI and NIST-LRI.

**Table 3 foods-14-00271-t003:** The r-OAVs of aroma compounds identified in 16 different varieties of Yunnan high-aroma white tea.

		r-OAVs															
Odor Threshold mg/kg	Compound Name	SS 1	SS 2	SS 3	SS 4	SS 5	SS 6	SS 7	NSS1	NSS2	NSS3	NSS4	NSS5	NSS6	NSS7	OUS	OU C
0.0058	2-Pentyl furan	99	42	109	122	98	33	36	95	107	150	7	56	28	13	121	431
0.0011	3-Methylbutanal	119	40	57	27	27	31	61	10	97	27	<1	22	<1	10	26	74
0.012	Pentanal	<1	3	4	4	5	3	3	8	<1	8	2	6	2	2	9	24
0.0024	Hexanal	13	103	259	299	328	207	156	554	517	400	8	453	4	123	725	1088
0.00021	(Z)-3-hexenal	<1	<1	<1	<1	<1	<1	<1	<1	353	<1	<1	386	<1	<1	426	<1
0.0028	Heptanal	<1	28	<1	<1	<1	34	28	54	43	<1	<1	49	24	16	74	<1
0.11	(E)-2-hexenal	18	<1	<1	8	12	<1	<1	<1	<1	<1	<1	<1	<1	<1	<1	36
0.03	2-Hexenal	<1	14	34	<1	<1	41	27	67	65	39	<1	54	32	23	128	<1
0.000025	(Z)-4-heptenal	1149	<1	<1	<1	465	<1	401	<1	679	757	<1	<1	<1	<1	<1	<1
0.000587	Octanal	178	<1	54	79	102	76	80	111	<1	148	91	<1	66	<1	<1	407
0.0011	Nonanal	447	248	360	393	334	543	237	336	343	334	203	264	343	189	941	1455
0.003	(E)-2-octenal	<1	6	16	13	26	11	18	13	9	44	<1	6	7	5	36	77
0.003	Decanal	18	<1	<1	<1	<1	160	<1	11	<1	14	<1	<1	<1	<1	<1	53
0.024	Benzaldehyde	27	16	<1	38	59	673	16	27	<1	52	3	12	26	7	<1	141
0.00019	(E)-2-nonenal	108	73	147	107	108	1	89	32	1	99	13	68	30	42	170	11
0.0008	(E,Z)-2,6-nonadienal	286	179	360	355	368	<1	249	243	<1	<1	<1	85	211	107	<1	853
0.003	Beta-cyclocitral	30	14	36	46	40	66	19	33	42	35	<1	21	38	12	169	183
0.0003	(E)-2-decenal	118	33	56	81	<1	<1	<1	<1	81	124	15	87	29	32	<1	561
0.0063	Benzeneacetaldehyde	<1	<1	<1	<1	<1	<1	<1	<1	<1	85	42	<1	<1	<1	<1	<1
0.053	Neral	189	188	17	7	229	42	350	3	102	139	9	126	86	114	389	671
0.004	3-Methyl-1-butanol	4	<1	10	<1	<1	<1	<1	<1	8	5	<1	<1	2	<1	<1	<1
0.065235	2-Heptanol	9	10	11	12	13	12	10	7	6	11	2	12	8	5	9	34
0.0056	1-Hexanol	35	75	66	40	73	108	61	66	56	43	3	31	36	52	37	205
0.0039	(Z)-3-hexen-1-ol	30	37	40	<1	<1	3	35	32	24	17	<1	37	35	<1	26	<1
0.1	(E)-2-hexen-1-ol	<1	<1	<1	<1	1	<1	5	1	1	<1	<1	1	1	2	<1	<1
0.0078	2-Octanol	3	3	<1	<1	7	<1	<1	<1	<1	6	1	4	<1	<1	<1	18
0.1	(Z)-linalool oxide (furanoid)	20	71	<1	34	19	<1	36	<1	21	21	1	19	12	23	<1	46
0.0015	1-Octen-3-ol	349	214	258	491	486	50	232	550	561	568	280	582	315	101	387	1704
0.0054	1-Heptanol	12	6	7	9	15	11	<1	<1	8	<1	9	11	21	4	14	42
0.19	(E)-linalool oxide (furanoid)	21	<1	33	36	18	<1	38	21	16	21	1	22	17	24	26	59
0.058	2-Nonanol	1	<1	<1	1	1	<1	1	<1	<1	2	<1	1	1	<1	1	4
0.00022	Linalool	147533	74350	191223	166244	124867	95	68931	136566	171089	125706	81166	210304	60232	60327	188436	448964
0.1258	1-Octanol	<1	<1	<1	<1	2	<1	1	<1	<1	2	<1	<1	1	<1	<1	7
0.0046	α-Terpineol	<1	<1	<1	<1	216	<1	<1	<1	323	103	<1	182	<1	<1	<1	569
0.062	Citronellol	<1	8	<1	<1	<1	<1	<1	4	4	2	<1	<1	<1	<1	11	5
0.68	Nerol	<1	1	<1	<1	<1	<1	<1	<1	<1	<1	<1	<1	<1	<1	2	2
0.0066	Geraniol	755	912	1170	565	271	161	260	569	670	546	420	1048	561	297	2508	1979
0.1	Benzyl alcohol	8	5	12	7	<1	16	5	3	4	7	<1	7	3	5	16	18
0.000015	Phenylethyl Alcohol	78858	93948	174859	<1	65216	50924	73270	27382	36134	75726	13098	4389	37308	58583	155039	235683
0.023	1-Penten-3-one	4	<1	<1	3	3	<1	<1	4	<1	<1	<1	<1	<1	<1	<1	4
0.000002	(E)-beta-damascenone	<1	39735	60750	<1	<1	<1	<1	11450	15160	37290	<1	<1	5360	20490	63835	47595
0.00378	α-Ionone	53	<1	34	56	<1	<1	<1	41	<1	<1	2	38	17	<1	76	247
0.012	3-Methylbutyric acid	1	3	<1	<1	<1	<1	<1	1	<1	<1	<1	<1	<1	1	2	5
0.036	Hexanoic acid	16	2	6	<1	17	194	3	<1	<1	<1	<1	11	6	4	35	51
0.00052	Methyl benzoate	<1	35	<1	47	58	<1	<1	<1	<1	<1	<1	<1	30	<1	<1	<1
0.04	Methyl salicylate	223	300	273	261	244	20	85	95	116	184	17	233	244	81	759	507
0.024	Toluene	<1	2	4	<1	<1	<1	<1	<1	4	4	3	<1	1	<1	6	13
0.0024	Ethylbenzene	<1	<1	<1	<1	<1	<1	24	<1	<1	7	<1	<1	<1	<1	<1	43
0.0012	β-Myrcene	832	624	792	754	342	659	355	878	817	487	1090	731	358	282	1047	2046
0.04	α-Phellandrene	<1	1	1	<1	<1	<1	<1	<1	1	1	<1	<1	<1	<1	1	6
0.034	Limonene	17	30	30	<1	8	11	<1	23	<1	<1	1	25	1	<1	<1	38
0.034	trans-β-Ocimene	7	7	22	<1	<1	3	<1	<1	9	5	21	<1	<1	2	1	14
0.065	γ-Terpinene	<1	5	1	<1	<1	<1	<1	<1	1	1	2	1	<1	<1	<1	1
0.0036	Styrene	<1	3	4	<1	<1	50	<1	7	9	7	3	<1	2	<1	<1	88

## Data Availability

The original contributions presented in the study are included in the article/Appendix A, further inquiries can be directed to the corresponding authors.

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
