# Peer review of "Comprehensive Characterization of the Odor-Active Compounds in Different Processed Varieties of Yunnan White Tea (Camellia sinensis) by GC×GC-O-MS and Chemometrics"

_foods, 2025, doi:10.3390/foods14020271_

Round 1

Reviewer 1 Report

Comments and Suggestions for Authors

The paper explores the aroma profile of white tea varieties. White tea samples were screened for 154 aroma compounds by GC×GC-O-MS. While the manuscript is generally interesting and holds sufficient interest for Foods, it has several limitations. I recommend a major revision of the text. The main disadvantage is the insufficient review of the literature, as well as the inadequate discussion of the results concerning other authors' research. It is necessary to conduct a more detailed literature review and compare the obtained results with previous research. I hope my comments will contribute to improving the presentation of your results.

Manuscript should been 'spell checked' and 'grammar checked'.

Major issues:

1.       Insufficient review of previous research, with only 25 references. The introduction section needs to be improved and expanded.

2.       The figures are too small, making it difficult to see what is depicted clearly. It is necessary to label parts of some figures, for example, to identify the samples grouped in the PCA analysis, so that the figure itself is self-explanatory (also Figures 5a, 5d). In many cases, it seems that the figures are not adequately discussed.

3.       The discussion of the results needs to be improved by comparing them with other studies and concluding why these specific results were obtained. While the authors have done this in a few cases, it is necessary to expand this part of the discussion. Additionally, I suggest that such discussions should not be limited to the same sentence structure, such as: "Wang et al. investigated that..."

Please check the following comments:

Abstract

The abstract is too long; according to the author’s guidelines, the limit is 200 words. The abstract should summarize the article's main findings; and indicate the main conclusions or interpretations.

Keywords

The keywords are not good, as they repeat words from the title.

Introduction

Lines 52-54: When writing scientific papers, it is not advisable to use sites like Wikipedia. It is better to use scientific papers that have been published in relevant scientific journals and have undergone proper peer review.

Lines 59-70: It seems that this paragraph is not well written; it appears disjointed. The authors first state that HS-SPME combined with GC × GC-O-MS is a significant technique for identifying trace components in teas. After that, they list what other authors have done. This way of discussing is not up to the standards of the journal Foods. The passage needs to be reformulated to highlight the most significant aromatic components of white tea, followed by the main findings of other authors. I emphasize again that the findings of other authors should be presented as a cohesive paragraph, rather than listing "Chan et al. did this," "Mu et al. investigated that," and so on.

Line 75: Please provide the number of samples that were collected and analyzed.

Material and methods

Line 131: A sentence cannot start with a number.

Line 141: Did the pre-screening test involve distinguishing between slight, medium, and intense concentrations for each descriptor? How many participants were allowed to participate in the final sensory evaluation?

Line 169: A sentence cannot start with a number.

Results and Discussion

Line 317: “Results revealed that the shaken, steam cooked…”

Line 299 – 333: In this section, obtained results should be compared and discussed in relation to other similar studies. I suggest authors see studies such as https://doi.org/10.1016/j.foodres.2023.113515.

Line 334: Figure dimensions are too small.

Line 375: Figure 5 cannot be mentioned before Figure 4.

Line 377-382: Figures 3a, 3b, and 3c are unclear and too small. The resolution of the figures needs to be improved, and I suggest moving these figures to the Supplementary file.

Line 450: Reference is missing.

Line 519: It is necessary to better discuss what is shown in Figure 4 and precisely explain what each segment of the figure represents.

Line 526: Use only full or abbreviated names for figures.

Line 587: Figure 5 is unclear. Why are there 3 heat maps in this figure, and what do they represent? Additionally, this figure contains too many small parts. It would be better to divide it into several figures.

Line 587: Please explain what is presented in Figure 4b.

References:

Reference number 3: The first letter of the journal is lowercase, but it should be uppercase. Also, all references should have an Abbreviated Journal Name. Please see the instructions for Authors for Journal Foods.

Author Response

Please have a check on the attached file.

Reviewer 2 Report

Comments and Suggestions for Authors

The manuscript entitled “Comprehensive characterization of the odor-active compounds in different processed varieties of Yunnan white tea (Camellia sinensis) by GC × GC-O-MS and chemometrics” describes the use of multiple dimensions, in terms of chromatographic, olfactometric, and spectrometric, coupled with supervised machine learning techniques to investigate the processed varieties of white tea information using their volatile profile.

The manuscript looks interesting, however, at the current state, a major revision is needed.

I think that a reorganization of the manuscript is necessary.

First, almost all the citations are from Chinese Scientists. It is evident that different varieties of white tea investigated in this study are from China and thus the citations related to the samples investigated are mainly from this part of the world, but the same cannot be for the analytical techniques applied. References should be the most important cutting-edge publications regarding sampling, VOCs analysis, SPME, comprehensive 2D-GC coupled with both O and MS, LRI, machine learning techniques etc, not just applications where these techniques were used. Please, modify the citations accordingly.

The authors should report in table 2 also the LRI in the NIST database and the ± differences compared to the LRI calculated.

What is the rationality to use a polar column in first dimension and a medium polar in the second? Moreover, no information regarding the modulator (cryogenic?) and the modulation time. Please, more details are necessary.

The use of alkanes for the calculation of LRI in a polar column may fail over time due to the normal degradation of the column during its use. Since several samples were analyzed, what did the authors do to check phenomena?

More details regarding the semi-quantification of using internal standard should be reported. This approach is more a normalization than a semi-quantification. The degree of ionization of analytes belonging to different chemical classes can be wildly different under EI-MS at 70 eV. In other words, analytes at the same concentration in the sample can show different intensities at the EI-MS detector. In my opinion, to semi-quantify the authors should use at least one internal standard per chemical class and when it is no possible use the IS that has the most similar chemical structure to the analytes investigated. Moreover, according to the text, I think there is a typo in the formula at line 252.

The authors reported compounds at ppb level (e.g. nitrogen-containing compound 20.6 ug/kg). Did the authors calculate the LOD and LOQ of the method?

How did the authors align the different chromatograms? Did they use a manual alignment or by software? Did they remove contaminants and artefacts before alignment (e.g., siloxanes)?

PLS-DA and OPLS-DA, if used without a statistical validation, can overfit the result forcing the classification. Did the authors validate the model? If yes, which kind of validation was applied?

Since the manuscript il long and not always easy to read, I suggest the authors to create a figure where the flow of the manuscript is described (from sample preparation to data elaboration).

Author Response

(The authors gave the same response as above.)

Reviewer 3 Report

Comments and Suggestions for Authors

The authors in this manuscript performed an evaluation of some differently processed white teas (camelia sinensis). The approach involed the use of multiple techniqe with the aim ot obtain a "comprehensive" information. This approach is by no means new, and a similar article has been recently published in foods (as also reported by the authors, note 16). The most notable difference form the previous article is the differing processing of the analysed teas, that leads to differing and perceivable nuances. For this reason while not completely novel, in the approach, this article stil gives more information on the subbject. I suggest a more thorough and refined bibligraphic research, avoiding to citate non scientific literature (wikipedia line 52).

I also found the pictures resolution not always adequate (probaly due to the resizing process of the software - likely centesimation) and the supplementary information table a little messy. Thisnotwithstanding I could reccomend the pubblication on foods, once revised these minor drawbacks. 

Round 2

Reviewer 1 Report

Comments and Suggestions for Authors

I am satisfied with the authors' corrections and have no further objections.

Author Response

Thank you.

Reviewer 2 Report

Comments and Suggestions for Authors

The authors made a big effort to improve the quality of the manuscript after revision. However, in my opinion, this is not enough for publication yet.

Table 2 reports the VOC identification based on LRI and MS%. However, in more case the differences between LRI calculated and NIST-LRI is too much (> +/- 30) making questionable the identification. Moreover, the LRI calculated have always a smaller value compared than the NIST-LRI. How do the authors explain this?

I think the the identification of VOCs has to have more strict criteria. Please, delate molecules from table 2 that do not meet these criteria. What is the cutoff value for LRI (e.g. +/- 30)? The delta-LRI between LRI calculated/NIST should be also reported in table 2. 

Author Response

Please have a check on the attached file. If more are needed, please let me know.
